# Boost Your Brainpower: 24 Daily Sleep Hacks for Active Lifestyles

Gian Mario Migliaccio [1,2,†], Gloria Di Filippo [3,†], Federica Sancassiani [4], Johnny Padulo [5,‡] and Luca Russo [6,*,‡]

1   Department of Human Sciences and Promotion of the Quality of Life, San Raffaele Rome Open University, 00166 Rome, Italy; gianmario.migliaccio@uniroma5.it
2   Maxima Performa, Athlete Physiology, Psychology and Nutrition Unit, 20126 Milan, Italy
3   Department of Psychology, Niccolò Cusano University, 00166 Rome, Italy; gloria.difilippo@unicusano.it
4   Department of Medical Sciences and Public Health, University of Cagliari, 09124 Cagliari, Italy; federicasancassiani@yahoo.it
5   Department of Biomedical Sciences for Health, Università degli Studi di Milano, 20133 Milan, Italy; johnny.padulo@unimi.it
6   eCampus University, 22060 Novedrate, Italy
*   Correspondence: luca.russo2@uniecampus.it
†   These authors contributed equally to this work.
‡   These authors contributed equally to this work.

**Abstract:** Sleep is a fundamental biological process that plays a pivotal role in the health and performance of physically active individuals (PAI). Sleep deprivation or poor sleep quality can negatively impact recovery capacity, concentration, coordination, and muscular strength, thereby compromising physical performance and increasing the risk of injuries. Objectives: This narrative literature review aims to examine the scientific evidence on the importance of sleep hygiene for the health and performance of PAI. A search was conducted for studies published on PubMed, Scopus, and Web of Science. Studies that investigated the effect of sleep hygiene on health and performance variables in athletes were included. The literature analysis highlighted that good sleep hygiene, adequate sleep duration (7–9 h per night), high sleep quality, and a regular sleep routine are associated with a range of benefits for the health and performance of PAI, including: (1) improved post-training recovery; (2) reduced risk of injuries; (3) enhanced concentration and attention; (4) improved coordination and muscle strength; (5) better mood and mental well-being; (6) reduced risk of chronic diseases. Sleep hygiene is a key factor for the health and performance of PAI. Implementing a comprehensive and personalized sleep hygiene routine can lead to significant improvements in the quality and quantity of sleep, with positive effects on physical and mental health, and overall well-being of PAI.

**Keywords:** sleep hygiene; physical activity; training; health; recovery; injuries; concentration; coordination; muscle strength; mood; mental well-being; chronic diseases

## 1. Introduction

Achieving high-level international objectives for athletes as well as conducting a physically active everyday life matching work, family, and training for physically active individuals (PAI) require both athletes and PAI to maintain a high standard of recovery, which allows for the maximization of adaptations from the substantial volume of training they undergo [1]. In this context, the role of sleep in maintaining, supporting, and optimizing cognitive and physical performance during and after training sessions is paramount [2]. However, athletes as well as PAI in the most common situation of everyday life are highly susceptible to sleep inadequacies, characterized by habitually short sleep of less than 7 h per night, along with poor sleep quality, primarily due to high fragmentation [3]. A study conducted with professional athletes observed lower sleep quality and sleep hygiene compared

to a peer age group, and difficulty falling asleep post-competition [4]. Under this light, sleep hygiene procedures must be considered fundamental. Sleep hygiene encompasses a series of behavioral and environmental practices designed to optimize sleep quality, which can significantly influence athletic performance. Originally developed for the treatment of insomnia, sleep hygiene has found significant applications in the sports context to optimize athletic performance [5]. During sleep hygiene training, PAI can be educated on a series of optimal practices, such as avoiding stimulants, implementing physical exercise routines, and creating a favorable sleep environment [6]. Sleep disorders represent an increasing area of concern in sports and exercise science. In this context, 'disorders' encompass a spectrum of sleep-related disturbances, with the six main ones highlighted as insomnia, circadian rhythm disorders, sleep-related breathing disorders, hypersomnia/narcolepsy, parasomnias, and restless legs syndrome/periodic limb movement disorder [7,8].

However, within athlete and PAI populations, other parameters have been incorporated into the definitions of sleep disorders, including prolonged sleep onset latency, excessive wakefulness after sleep onset, short total sleep time, low sleep efficiency, or poor sleep quality based on subjective and/or objective assessments, which can affect general health, injury risk, and career longevity [9]. Athletes and PAI may suffer from these conditions and might not be aware of them unless they are specifically assessed and enrolled in a sleep hygiene program. Sleep hygiene education is considered an economical and readily accessible lifestyle intervention that can limit sleep deprivation, thereby maximizing performance in the athlete population [6].

The implementation of sleep hygiene practices does not necessarily require direct supervision by a sports medicine physician, potentially increasing its accessibility for physically active individuals (PAI) who may be reluctant to seek medical intervention. Information on sleep hygiene can be easily disseminated through various channels, such as print materials or online platforms. However, it is important to note that PAI with undiagnosed sleep disorders might misuse sleep hygiene practices such as going to sleep late due to training in the late evening or waking up very early in the morning to train or not following the guidelines for correct physical activity [10,11]. In such cases, it is crucial to direct them towards treatments that are more specific.

Sleep deprivation is a widespread issue among PAI, who often fail to achieve the recommended 7–9 h of sleep per night, achieving a lesser amount compared to non-PAI [12]. This trend has been observed across various types of sports, both individual and team, and in both strength and endurance disciplines.

A comprehensive evaluation and expansion of the current evidence base supporting sleep hygiene recommendations in the context of physical exercise are necessary to elucidate their efficacy, particularly considering the different phases of the 24 h cycle: before sleep, during sleep, and after sleep. With a particular focus on application in PAI, this review aims to focus on identifying conceptual and methodological areas for direct implementation into a 24 h cycle routine that fosters behaviors conducive to improving both the quantity and quality of sleep in PAI. This includes specific strategies for each of the three phases, such as relaxation techniques and reducing blue light exposure before sleep, maintaining an optimal sleep environment throughout the night, and waking practices that facilitate the transition from sleep to full activity.

The goal will be reached by examining critically the empirical evidence for the individual components of sleep hygiene recommendations, identifying the issues caused by sleep deprivation, with a particular focus on variations in behaviors and needs related to the three distinct phases of the sleep–wake cycle: the pre-sleep, during-sleep, and post-sleep phases. This literature review aims to identify knowledge gaps in our current understanding of sleep hygiene recommendations within the physical exercise context and propose a structured sleep periodization framework that takes into account the diverse needs related to the three phases of the sleep–wake cycle. This periodized approach aims to optimize physical performance through improved sleep management, adapting sleep hygiene strategies to circadian rhythms and the specific needs of PAI at various stages of their daily routine.

## 2. Materials and Methods

For the review, we conducted bibliographic searches on PubMed/Medline to identify studies published from 2000 to 2024, using MeSH keywords such as "Sleep Deprivation", "Athletes", "Sleep Hygiene", "Physical Exercise", and "Athletic Performance", along with additional terms like "sleep extension", "sleep quality", "jet lag", and "circadian rhythm", in English, without limitations related to gender or age. We also manually examined the bibliographies of selected articles to verify their relevance. Studies that did not involve athletes or were focused on individuals with sleep-related pathologies, as well as those that did not consider sleep as the main outcome measure or that were animal studies, were excluded from our analysis. The search yielded two hundred and nine studies, of which one hundred and one were deemed relevant and thus subject to further examination. The selected studies presented a great heterogeneity, preventing an aggregated synthesis of the data. Most were small-scale studies with ≤20 participants or small cohort groups; hence, a descriptive review was opted for. The PAI in the studies participated in endurance disciplines, strength sports, and activities with combined physiology and were both men and women.

## 3. Pre-Sleep Phase

The initial phase of sleep is a crucial element for physical performance and can be a significant source of emotional stress for PAI [13,14]. While some individuals among the younger population can fall asleep quickly, with the time required increasing with age from 2 to 20 min [15], PAI may encounter considerable difficulties in reaching the sleep state, taking a significant amount of time to enter the REM phase due to an increase in epinephrine following intense exercise [16]. For PAI, the results demonstrate an increase in slow-wave sleep, a reduction in REM sleep, and a prolonged REM sleep latency, especially pronounced in the first 3 h after sleep onset. These changes in sleep architecture are associated with increased morning energy levels and perceived sleep quality. One proposed mechanism for the delayed REM sleep is increased aminergic neurotransmitter levels and sympathetic activity, supported by the observation that athletes with higher norepinephrine levels exhibited greater REM latency following intense exercise [17]. This issue can generate a negative psychological impact, such as frustration, and have adverse consequences on athletic performance the following day [18].

However, the adoption of specific behavioral precautions and the modification of certain habits can facilitate the process of falling asleep, placing greater emphasis on factors often overlooked [19]. A common mistake is the tendency to underestimate the efficacy of behavioral interventions, opting instead for pharmacological or external solutions [20]. This approach, although it may offer temporary relief, does not address the underlying cause of the problem and can lead to side effects. After the treatment ends, individuals tend to return to their initial state, which is often related to a misalignment between the circadian rhythm and unhealthy lifestyle habits [21].

### 3.1. Adherence to the Circadian Rhythm

The circadian rhythm, a biological cycle of approximately 24 h, regulates a series of physiological, cognitive, and behavioral changes in the organism [22]. One of the primary functions of this rhythm is to prepare the body for a state conducive to sleep [23]. This state is modulated by the endogenous biological clock, responsible for the secretion of specific hormones that induce states of sleep or wakefulness. Athletes should maintain a consistent sleep schedule by synchronizing their body with the sleep phase, facilitating the achievement of an optimal state of sleep at a predefined time, favored by the correct sleep pressure. However, there is no universal sleep schedule; congruence with one's own chronotype can be assessed through the observation of spontaneous awakening in the absence of external stimuli such as an alarm clock [24].

In the case of a chronotype misalignment, which could compromise daily commitments such as school or work, the most effective strategy is simply to advance the bed-

time [25]. Given the rigid structure of social and sporting commitments, the waking time is often fixed; therefore, the only margin for maneuver to align sleep with the chronotype is to modify the bedtime. This can be carried out gradually, by advancing the sleep time by 5–10 min a week [26].

### 3.2. Promotion of a Dark Environment

Ambient lighting has a significant impact on the human circadian rhythm through the activation of specific photoreceptors which transmit signals to the central nervous system via the retina and the optic nerve [27]. These photoreceptors, divided into cones and rods, remain functional regardless of the state of the eyelids [28]. The rods, in particular, exhibit a sensitivity to light up to 4000 times higher than that of the cones, enabling orientation in low-light conditions [29].

The entry of light through the eyelids can generate ambiguous signals to the brain, compromising the regulation of melatonin and sleep, and, consequently, compromising its effectiveness in facilitating sleep [30]. Light exposure, even at low intensities such as 3 lux, can have implications for sleep quality and circadian rhythmicity. Recent human studies have demonstrated that exposure to light at night levels of 5–10 lux during sleep may influence sleep physiology and daytime brain function, with effects on mood and depression susceptibility [31]. Notably, light at night intensities around 3 lux are achievable not only from installed emergency lighting in public spaces but also from display screens of clocks, alarms, televisions, and electronic devices. Therefore, even minimal light sources in the sleeping environment, including electronic displays, can potentially disrupt sleep quality and biological rhythmicity.

These findings underscore the importance of minimizing light exposure during sleep, particularly from artificial sources, to maintain optimal sleep–wake cycles and associated physiological processes. Maintaining a sleep environment that is as dark as possible is therefore crucial to ensure optimal melatonin release.

### 3.3. Airplane Mode

Deactivating electronic devices such as televisions, tablets, and smartphones before nighttime rest is a practice recommended for improving sleep quality [32]. This advice, which might seem like mere parental guidance, is actually supported by robust scientific evidence. Prolonged interaction with tablets or smartphones, exceeding 8 h per day or even just for 30 min before the rest period, can have a negative impact on subsequent sleep [33]. However, simply turning off the device's screen may not be sufficient. It is crucial to place the device at a significant distance from the resting place to optimize sleep quality [34]. For example, it is suggested that electronic devices like televisions should be positioned at least 6 feet (2 m) away from the sleeping area [35]. Merely, activating "airplane mode" may not be adequate; it is more effective to completely turn off the device or place it in silent mode at a reasonable distance from the bed. Even in the case of a device with alerting properties, it can be suggested to position it far from the bed.

### 3.4. Limitation of Technology

The practice of reading can have a positive impact on sleep quality, as long as it is not carried out on electronic devices such as smartphones or tablets [36]. Reading traditional books, preferably under yellow-toned light, can improve sleep quality by up to 22% compared to the sleep quality of non-readers [37]. Additionally, reading in environments with dim lighting can double these benefits. However, it is important to note that the genre of reading can affect sleep quality; individuals who are emotionally reactive to certain stories might experience a delay in falling asleep [38]. E-books without backlighting might have a lesser impact, but the emission of LED lights can alter melatonin secretion, potentially affecting circadian rhythms and cognitive performance [39].

### 3.5. Training Schedules

It is noteworthy that physical activity, both intense and moderate, has a positive effect on sleep quality [40]. However, for PAI engaging in intense physical activities, it is advisable to conclude workouts at least two hours before sleep to avoid negative interferences with sleep quality [41]. Furthermore, the intense and white lighting typical of sports fields and gyms can inhibit the release of melatonin, further compromising sleep quality [42].

### 3.6. Stress Reduction

Falling asleep can be hindered by high mental activity, including anxious thoughts and worries [43]. In the prehistoric era, the sympathetic nervous system's response was activated only in the presence of immediate threats, while the hectic pace of modern life and constant exposure to variable stimuli maintain a high chronic activation of the sympathetic system, leading to elevated stress levels and sleep disorders [44]. Mindfulness and meditation techniques have shown potential benefits in improving sleep quality, although the scientific evidence is still limited [45].

### 3.7. Activation of the Parasympathetic System

For the improvement of sleep quality, breathing directly influences the ease of falling asleep, the reduction in the frequency of nocturnal awakenings, and the optimization of sleep efficiency [46]. These benefits have been observed in subjects, including those suffering from insomnia, who practiced slow and rhythmic breathing exercises for a period of 20 min before sleep [47]. Slow and deep breathing, increasingly supported by scientific evidence, involves a respiratory rate that varies between 4 and 10 breaths per minute (0.07–0.16 Hz), as opposed to the typical 10–20 breaths per minute (0.16–0.33 Hz) of the average human, which can have a close relationship with athletic performance [48].

This breathing mode, when combined with proper sleep hygiene and relaxation techniques, has proven particularly effective in facilitating falling asleep and re-sleeping, quickly synchronizing the heart rate with the respiratory rhythm [49]. Breathing techniques that maintain a rhythm of 0.1 Hz, corresponding to about six breaths per minute, have shown promising results in activating the parasympathetic nervous system, which acts in opposition to the sympathetic system, thereby promoting physical relaxation, better emotional control, and psychological well-being [50].

### 3.8. Maintenance of Sleep Regularity

The regularity of the circadian rhythm is a crucial element for maintaining quality sleep and, consequently, for overall health [21]. Although our ancestors relied on natural cycles of light and darkness to regulate sleep, modern life has introduced a number of factors that can disrupt this natural rhythm. For example, going to bed at different times each night can cause a form of "social jet-lag", with negative effects on both sleep quality and overall health [51].

This irregular behavior can lead to chronic sleep delay and increased variability, both associated with negative health outcomes; therefore, establishing a regular nighttime routine is imperative.

### 3.9. Body Weight Management

To optimize sleep quality, body weight and dietary habits can have a significant impact on sleep quality. For instance, a heavy meal consumed shortly before going to bed can disrupt sleep cycles and compromise rest quality [52]. This assertion is supported by research from Crispim et al. (2007), who demonstrated that late-night high-fat meals can negatively impact sleep quality and architecture [53]. Some foods, such as refined carbohydrates, may have a sedative effect but can also cause a glycemic spike, followed by an insulin response and potential nocturnal hunger, which could lead to nighttime awakenings [54].

This effect has been demonstrated in a study by Afaghi et al. (2007), which found that high-glycemic-index carbohydrate meals consumed 4 h before bedtime decreased sleep onset latency [55]. An ideal diet to promote quality sleep should include whole grains, nuts, dairy, fruits, and vegetables [56]. Furthermore, St-Onge et al. (2016) observed that low fiber intake and high saturated fat and sugar intake were associated with lighter, less restorative sleep with more arousals [57]. Additionally, it is important to consider that digesting a meal can take 2 to 3 h, during which sleep may be compromised [58]; Ormsbee et al. (2016) observed that protein consumption 30 min before bedtime increased overnight metabolism, suggesting prolonged digestive activity [59]. Going to bed during the digestion process might cause discomfort or nausea and further slow down digestion [60]. Kinsey and Ormsbee (2015) reviewed the effects of nighttime eating on sleep, appetite, and metabolism, providing further evidence for the complex relationship between late-night eating and sleep quality [61].

### 3.10. Impact of Music on Sleep Quality

Music has been identified as a non-pharmacological tool for improving sleep quality [62]. Nilsson (2009) observed a significant reduction in cortisol levels in patients listening to relaxing music [63], while Salimpoor et al. (2011) found that pleasurable music triggers dopamine release in the brain [64]. Several studies have demonstrated that music can reduce cortisol levels, the stress hormone, and promote the release of dopamine, a neurotransmitter associated with pleasure [65]. Lai and Li (2011) demonstrated that older adults listening to relaxing music at bedtime for 45 min over 3 weeks showed significant improvements in sleep quality, including longer sleep duration and less sleep disturbance [66]. These effects can contribute to better quality sleep by reducing alertness and enhancing feelings of well-being. Additionally, music can have a calming effect on the autonomic nervous system, leading to slower breathing, a lower heart rate, and reduced blood pressure [67].

There is no unanimous consensus on the type of music most effective for improving sleep quality; however, it has been suggested that selecting tracks with a beats per minute (BPM) between 60 and 80 could be particularly effective as it may synchronize with the resting heart rate [68]. Huang et al. (2016) found that music with a tempo of 60–80 BPM was most effective in reducing anxiety and improving sleep quality in patients [69]. At the same time, it is worth noting that familiar and repetitive music can trigger involuntary musical imagery that worsens sleep quality [70]. Scullin et al. (2021) found that individuals who frequently listen to music are more likely to experience 'earworms' at night, which can interfere with sleep quality [70].

### 3.11. Bath or Hot Shower

Pre-sleep bathing or showering can positively impact sleep, particularly in athletes [71]. A meta-analysis by Haghayegh et al. (2019) demonstrated that bathing 1–2 h before bedtime in water of 40–42.5 °C significantly improves sleep quality and reduces sleep onset latency [72]. This effect is attributed to heat loss induced by vasodilation [72].

Furthermore, maintaining warm feet during sleep in a cool environment through the use of bed socks has been associated with shortened sleep onset latency, increased total sleep time, and reduced awakenings throughout the night [73,74]. Ko and Lee (2018) found that wearing bed socks significantly enhanced sleep quality and reduced sleep onset time in adults sleeping in cool conditions [73]. A meta-analysis by Jiang et al. (2024) further corroborated that foot thermal therapy can improve various aspects of sleep quality, particularly in older adults [74].

### 3.12. Melatonin, Herbs, and Tryptophan

Melatonin is known as the "sleep hormone" and has various applications, including in the treatment of jet lag and some sleep disorders [75]. A meta-analysis by Liira et al. (2014) found that melatonin slightly improves sleep quality, but the evidence is of low quality [76],

and the effectiveness of taking exogenous melatonin in shift workers is still a subject of debate [77].

Herbs such as Valerian and Chamomile are often used as sleep aids, but scientific research has not yet provided solid evidence of their effectiveness [78]. The systematic review by Shinjyo et al. (2020) found limited evidence supporting Valerian's efficacy for sleep improvement [78]. Similarly, Hieu et al. (2019) conducted a meta-analysis on Chamomile's effects on sleep quality, showing modest benefits [79].

Other plants like passionflower, lavender, lemon balm, magnolia bark, and ashwagandha have shown potential, but results are mixed [80,81]. Herbal teas, however, have shown positive signs in improving sleep quality [82].

Tryptophan is an essential amino acid that acts as a precursor to serotonin, a neurotransmitter that regulates mood, sleep, and appetite [83]. A deficiency in tryptophan can lead to low levels of serotonin, causing conditions such as depression, anxiety, and insomnia [84]. Tryptophan is involved in the production of serotonin and melatonin, which regulate sleep–wake cycles [85]. Sutanto et al. (2022) conducted a meta-analysis demonstrating that tryptophan supplementation can improve sleep quality, particularly sleep latency and sleep efficiency [86].

To improve sleep quality, it is advisable to follow a diet rich in tryptophan. Foods such as tuna, salmon, chicken, beans, pumpkin seeds, almonds, bananas, yogurt, milk, and cheese are good sources of tryptophan [87]. The absorption of tryptophan is facilitated when consumed with complex carbohydrates [87].

### 3.13. Avoiding Alcohol

Contrary to popular belief, alcohol is not a good sleep inducer. While it may reduce sleep latency and increase NREM sleep in the first half of the night, in the second half, it suppresses REM sleep, leading to a "REM rebound" with altered dreaming [88–91]. Alcohol abstinence can also disrupt sleep homeostasis, increasing nighttime awakenings [90].

## 4. During-Sleep Phase

### 4.1. Environmental Temperature and Sleep

The environmental temperature of the sleeping area can significantly impact sleep quality. The National Sleep Foundation suggests a bedroom temperature of 16–19 °C (60–66 °F) to promote sleep [92]. Lowering body temperature can facilitate sleep induction and minimize energy expenditure [93].

### 4.2. Sleeping Positions

The position in which one sleeps can affect sleep quality and overall health. The National Sleep Foundation states that sleeping on one's back is generally best for health, but only 8% of people adopt this position [94].

Sleeping on the side is the most common position and can promote better airflow. It is often recommended for people who snore or have sleep-related breathing issues such as obstructive sleep apnea and can also alleviate neck and back pain [95].

Sleeping on the back can be comfortable for some people, especially those with back pain. However, it may intensify snoring and acid reflux [96].

Sleeping on the stomach is less common and can increase pressure on the spine, causing neck and back pain. It can also make breathing more difficult [96].

### 4.3. Mattress and Pillow

The comfort of the mattress and pillow is crucial for good sleep quality. A medium-firm mattress is generally recommended for comfort and spinal alignment [97]. Regarding pillows, the use of an orthopedic memory foam pillow is suggested to maintain a constant temperature and reduce neck pain related to sleep [98,99].

### 4.4. Electromagnetic Waves

Environmental electromagnetic fluctuations can affect sleep quality. Devices such as cell phones, Wi-Fi routers, and household appliances are often responsible for electromagnetic fields that can cause sleep disturbances and other health issues [100,101]. Alterations in electromagnetic fields can also affect the circadian rhythm [102].

### 4.5. Environmental Noise

Noise, especially in urban areas, can have a negative impact on sleep quality. It can cause physiological changes such as variations in heart rate and blood flow [103]. Eliminating environmental noise can improve sleep quality and reduce the latency of sleep onset [104].

### 4.6. Wearing an Eye Mask or Earplugs

Using an eye mask can have beneficial effects on memory, alertness, and cognitive function the next day even though it does not show immediate effects on sleep itself [105]. The use of earplugs can extend the duration of deep sleep (N3), especially for those who are not bothered by them [106].

### 4.7. Avoiding Interruptions

Nighttime interruptions leading to fragmentation can be caused by various factors, both physiological and pathological, resulting in increased objective sleepiness, decreased psychomotor performance in tasks involving short-term memory, increased reaction time, or vigilance [107]. For physiological causes, such as the need to drink or eat, it is advisable to regulate water and food intake throughout the day. For instance, limit fluid intake after 19:00 and avoid simple sugars at dinner to prevent glycemic spikes that can induce nocturnal hunger.

## 5. Post-Sleep Phase

### 5.1. Watching the Sun

Exposure to sunlight in the morning can have beneficial effects on our sleep–wake cycle by blocking residual melatonin production and activating cortisol [108]. This practice suppresses melatonin production during the day and increases cortisol levels, improving alertness, concentration, and efficiency in daily activities [108,109]. This is particularly useful for those who have experienced sleep deprivation as it helps to regulate the cycle of light and darkness exposure, improving cognitive performance and mood through the production of serotonin [110].

### 5.2. Not Sleeping during the Day

Avoiding prolonged afternoon sleep is crucial to maintaining a good sleep–wake cycle. A nap of over an hour can negate the "sleep pressure" accumulated during the day, making it difficult to fall asleep the following night and compromising sleep quality [111]. Instead, a "power nap" of about 20 min can provide an ideal balance, improving alertness without negatively affecting nighttime sleep [112]. Elite athletes show significantly shorter sleep latencies, suggesting that napping behavior may reflect the ability to fall asleep on demand rather than drowsiness resulting from sports-related sleep debts [111].

### 5.3. Engaging in Physical Exercise

Physical exercise is a powerful ally in improving sleep quality. Not only does it help reduce stress and anxiety, but it also contributes to regulating melatonin levels, the hormone that promotes sleep. Although not apparently specific for athletes, consistent activity of about 60 min four or five times a week has been shown to be effective even for patients with insomnia [113,114].

*5.4. Limiting Caffeine*

Caffeine is a stimulant that can interfere with sleep up to 8.8 h after consumption. To avoid sleep problems, it is recommended not to consume caffeine after 14:00–14.30 if going to bed at 23:00 [115]. Anyway, caffeine-sensitive people might need to discontinue caffeine consumption any time after noon.

## 6. Discussion

The literature review underlines that adherence to specific recommendations across the three phases of the sleep–wake cycle—the pre-sleep, during-sleep, and post-sleep phases—can significantly enhance both sleep quality and quantity. These improvements are associated with positive outcomes in the athletic performance, physical and mental health, and overall well-being of physically active individuals (PAI).

The pre-sleep phase involves several critical strategies for optimizing sleep conditions. Maintaining consistent sleep schedules aligns circadian rhythms with environmental cues, facilitating sleep initiation and quality enhancement. Minimizing exposure to blue light emissions from electronic devices and reducing technology use before bedtime are crucial for promoting melatonin production and sleep induction. Stress management techniques, including mindfulness, meditation, and controlled breathing exercises, are recommended to facilitate the transition to sleep. Furthermore, engaging in non-stimulating activities, such as reading physical books and listening to calming music, in conjunction with avoiding intense physical exercise proximal to bedtime, can contribute to improved sleep onset. Dietary considerations, weight management, and the judicious use of aromatherapy or supplements (e.g., melatonin and specific herbs) may also be beneficial, albeit under appropriate medical supervision.

The sleep environment is a critical factor in sleep quality. Optimal bedroom temperature (cool) and comfortable sleeping arrangements, including medium-firm mattresses and appropriate pillows, are essential for minimizing discomfort and pain. Reduction of electromagnetic exposure and environmental noise, coupled with the use of sleep aids such as eye masks and earplugs, can significantly enhance sleep quality by mitigating disturbances.

Post-sleep behaviors also play a crucial role in regulating sleep patterns and overall health. Morning exposure to natural light assists in resetting the circadian rhythm, while nutritious breakfast consumption and adequate hydration throughout the day support energy levels and improve subsequent sleep quality. Consistent physical activity during daytime hours further promotes both daytime alertness and nighttime sleep readiness.

In summary, our findings underscore the importance of comprehensive management of the sleep environment and sleep hygiene behaviors in optimizing the restorative effects of sleep, thereby enhancing physical readiness and performance. While this review primarily addresses physically active individuals, it is noteworthy that these practices may be beneficial for both sedentary and active adults. Finally, it is important to acknowledge that, in cases of severe or persistent sleep disturbances, a multifaceted approach incorporating pharmacological interventions, such as benzodiazepines, may be warranted [91]. However, a comprehensive discussion of pharmacological treatments is beyond the scope of this review.

## 7. Limitation

We acknowledge several limitations in our review. While the suggestions we have reported are applicable to a healthy population and athletes, we cannot definitively demonstrate that they can improve performance. This is an objective that future studies will need to address. We also found that the available data were not always able to provide answers that could demonstrate differences by sex and age. We recognize that a subsequent meta-analysis will be necessary to establish greater scientific consistency on this topic, and can also collect hygiene rules extended to 24 h. It is important to note that we found that the

scientific consistency regarding athletic performance for many of the provided suggestions is currently lacking.

## 8. Conclusions

Despite these limitations, we believe our comprehensive literature review on evidence-based protocols for sleep hygiene provides valuable insights for both athletes and the general population. The hygiene rules and suggestions we have outlined are based on current scientific evidence and offer a solid foundation for improving sleep quality. While the direct link to athletic performance enhancement requires further investigation, we emphasize the importance of good sleep hygiene for overall health and well-being. We intend this review to serve as a reference point for healthy individuals seeking to optimize their sleep patterns and potentially support their athletic endeavors. We suggest that future research should focus on addressing the identified gaps, particularly regarding age and gender differences, and establishing more concrete connections between sleep hygiene practices and athletic performance.

**Author Contributions:** Conceptualization, G.M.M.; methodology, L.R., G.D.F. and F.S.; software, J.P.; validation, G.M.M., J.P. and L.R.; formal analysis, L.R.; investigation, G.M.M.; resources, J.P.; data curation, L.R.; writing—original draft preparation, G.M.M., G.D.F. and F.S.; writing—review and editing, L.R.; visualization, G.M.M.; supervision, J.P.; project administration, G.M.M., J.P. and L.R. All authors have read and agreed to the published version of the manuscript.

**Funding:** This research received no external funding.

**Conflicts of Interest:** The authors declare no conflicts of interest.

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
