# Peer review of "Boost Your Brainpower: 24 Daily Sleep Hacks for Active Lifestyles"

_applsci, doi:10.3390/app14156701_

Round 1

Reviewer 1 Report

Comments and Suggestions for Authors

I thank you the editor for the invitation to review the manuscript. 

This is a very interesting and comprehensive literature review on evidenced-based protocols on sleep hygiene.

Contrary to what anticipated in the aim of the study, this paper does not touch on performance of athletes, but only provides sleep hygiene indications. I would change that on the aim of the study.

For the body of the paper, it is very well summarized and referenced with important quotes. I would suggest to add whether there are differences between females and males and changes with age.

I would also suggest that the 5.5. avoiding alcohol is moved to the pre-sleep stage phase, instead of the post-sleep stage.  

For the rest, this is a very well summarized paper and should be considered as a references for athletes but also general population. The authors correctly said that this is for healthy individuals, therefore disorders affecting sleep are not included. 

Author Response

I thank you the editor for the invitation to review the manuscript. 

This is a very interesting and comprehensive literature review on evidenced-based protocols on sleep hygiene.

Contrary to what anticipated in the aim of the study, this paper does not touch on performance of athletes, but only provides sleep hygiene indications. I would change that on the aim of the study.

R: Thank you to the reviewer for this suggestion. The queries we used (line 105) allowed us to identify articles that could also provide elements related to athletic performance. However, we agree that many of the suggestions provided currently lack scientific consistency regarding athletic performance, so we have included a specific recommendation and limitations to be more precise, accepting your suggestion.

For the body of the paper, it is very well summarized and referenced with important quotes. I would suggest to add whether there are differences between females and males and changes with age.

R: Thank you for this important suggestion. This was also one of our considerations during the writing phase, but unfortunately, the data were not always able to provide an answer that could demonstrate differences by sex and age. However, we believe this is an aspect that should be noted, and we have integrated it into the limitations.

I would also suggest that the 5.5. avoiding alcohol is moved to the pre-sleep stage phase, instead of the post-sleep stage.  

R: Your suggestion has been appreciated. Since alcohol also has an interference from the morning, we inserted it at point 5.5. However, upon rereading the references, the effect related to alcohol is greater when consumed in the hours preceding sleep, for example at dinner rather than at lunch. We have therefore accepted your suggestion and moved it to point 3.13.

For the rest, this is a very well summarized paper and should be considered as a references for athletes but also general population. The authors correctly said that this is for healthy individuals, therefore disorders affecting sleep are not included. 

R: Thank you, we have further modified the text in the conclusions to better highlight this suggestion that we have incorporated.

Reviewer 2 Report

Comments and Suggestions for Authors

This scientific review has a well-organized structure, providing a clear and comprehensive overview of factors influencing sleep quality and offering practical recommendations. However, it lacks detailed descriptions of the underlying molecular and physiological mechanisms, which are essential for understanding how these factors and interventions specifically impact sleep. Additionally, the narrative could benefit from more precise language and a critical evaluation of the varying efficacy of the recommended interventions, particularly in different populations such as athletes versus the general public. The review sometimes reads more like a magazine article than a scientific paper, with a tone and style that occasionally lack the precision and depth expected in academic writing.

Here's a detailed assessment of where the text falls short in terms of scientific support:

  • While it is a well-known fact that heavy meals can disrupt sleep, the text does not provide specific studies or data to support this claim. Including references to research on the impact of late-night eating on sleep would strengthen the argument.
  • The effects of refined carbohydrates on sleep are mentioned, but the text does not cite any studies that have investigated this. Referencing studies on how high-glycemic foods affect sleep would be beneficial.
  • The recommendation for a low GI diet to improve sleep is plausible but should be supported by scientific studies. For instance, research on how low GI diets influence sleep onset and duration would be appropriate.
  • The assertion that digesting a meal takes 2-3 hours and can impact sleep should be supported with references to research on digestion times and their effects on sleep quality.
  • The claims about music reducing cortisol and increasing dopamine should be backed by specific studies. Citing research that has measured these effects would enhance credibility.
  • The text correctly notes that music can have calming effects, but it needs references to studies that have investigated these physiological changes.
  • The suggestion to use music with a BPM of 60-80 is interesting but should be supported by research that has found this range to be effective for sleep improvement.
  • The potential negative impact of familiar and repetitive music on sleep is an intriguing point that should be substantiated with scientific studies.
  • The explanation of heat loss induced by vasodilation is scientifically sound but lacks specific references. Including studies on how pre-sleep baths or showers influence sleep onset would be beneficial.
  • The benefits of warm feet during sleep are mentioned, but the text should cite studies that have found these specific results.
  • While the general information about melatonin is accurate, the discussion on its effectiveness, particularly for shift workers, should be backed by studies that have investigated these claims.
  • The text mentions the use of herbs like Valerian and Chamomile but lacks references to scientific studies. Including research on the efficacy of these herbs in improving sleep would be beneficial.
  • The role of tryptophan in serotonin and melatonin production is correctly noted, but the text should reference studies that have examined the impact of tryptophan-rich diets on sleep quality.

Comments on the Quality of English Language

The English used in the review is generally correct and readable, with a clear and structured presentation of ideas. However, there are occasional awkward phrasings and minor grammatical errors that could be refined to enhance readability and coherence. For example, sentences could be made more concise, and transitions between sections could be smoother to improve the overall flow of the document.

Author Response

This scientific review has a well-organized structure, providing a clear and comprehensive overview of factors influencing sleep quality and offering practical recommendations. However, it lacks detailed descriptions of the underlying molecular and physiological mechanisms, which are essential for understanding how these factors and interventions specifically impact sleep. Additionally, the narrative could benefit from more precise language and a critical evaluation of the varying efficacy of the recommended interventions, particularly in different populations such as athletes versus the general public. The review sometimes reads more like a magazine article than a scientific paper, with a tone and style that occasionally lack the precision and depth expected in academic writing.

R: We thank the reviewer for this critical yet constructive comment. The choice to use language that may sometimes seem less scientific and more informative was part of our initial reflection where we considered our role and our contribution to scientific research. With this narrative review, we wanted to send a message to a world beyond the scientific community, trying to maintain a more fluid description that could be understood even by those who do not operate in our field. We ideally imagined our contribution to research as offering a practical yet consistent guide that could be realistically applied by the general population and athletes. This objective led to the choice of using this tone.

We have changed the tone of writing on lines 49,55,69,82 and 95 in order to follow your suggestions.

However, we share the consideration of better argumentation and offering greater depth, and we will try our best to satisfy the suggestions you have reported below. To further address your request to make the paper more scientific and less like a magazine article, we have added DOI (Digital Object Identifier) for each reference, enhancing the scientific rigor and allowing for easier verification of our sources.

Here's a detailed assessment of where the text falls short in terms of scientific support:

  1. While it is a well-known fact that heavy meals can disrupt sleep, the text does not provide specific studies or data to support this claim. Including references to research on the impact of late-night eating on sleep would strengthen the argument.
  2. The effects of refined carbohydrates on sleep are mentioned, but the text does not cite any studies that have investigated this. Referencing studies on how high-glycemic foods affect sleep would be beneficial.
  3. The recommendation for a low GI diet to improve sleep is plausible but should be supported by scientific studies. For instance, research on how low GI diets influence sleep onset and duration would be appropriate.
  4. The assertion that digesting a meal takes 2-3 hours and can impact sleep should be supported with references to research on digestion times and their effects on sleep quality.

R: Thank you for your insightful comments. We appreciate and concur with the reviewer's observations. In response, we have substantially revised the text to address these concerns. We have integrated new and appropriate references to support our statements about the effects of heavy meals, refined carbohydrates, digesting and low GI diets on sleep quality:

  1. Crispim CA, Zalcman I, Dáttilo M, et al. The influence of sleep and sleep loss upon food intake and metabolism. Nutrition Research Reviews. 2007;
  2. Kinsey and Ormsbee (2015) [2], who reviewed the effects of nighttime eating on sleep, appetite, and metabolism.
  3. St-Onge, M. P., Roberts, A., Shechter, A., & Choudhury, A. R. (2016). Fiber and saturated fat are associated with sleep arousals and slow wave sleep. Journal of Clinical Sleep Medicine, 12(1), 19-24.
  4. Ormsbee MJ, Gorman KA, Miller EA, Baur DA, Eckel LA, Contreras RJ, Panton LB, Spicer MT. Nighttime feeding likely alters morning metabolism but not exercise performance in female athletes. Appl Physiol Nutr Metab. 2016
  • The claims about music reducing cortisol and increasing dopamine should be backed by specific studies. Citing research that has measured these effects would enhance credibility.
  • The text correctly notes that music can have calming effects, but it needs references to studies that have investigated these physiological changes.
  • The suggestion to use music with a BPM of 60-80 is interesting but should be supported by research that has found this range to be effective for sleep improvement.
  • The potential negative impact of familiar and repetitive music on sleep is an intriguing point that should be substantiated with scientific studies.

R: We appreciate the reviewer's comments and have addressed them by incorporating additional scientific evidence:

  • Nilsson, U. (2009). Soothing music can increase oxytocin levels during bed rest after open-heart surgery: a randomised control trial. Journal of clinical nursing, 18(15), 2153-2161.
  • Salimpoor, V. N., Benovoy, M., Larcher, K., Dagher, A., & Zatorre, R. J. (2011). Anatomically distinct dopamine release during anticipation and experience of peak emotion to music. Nature neuroscience, 14(2), 257-262.
  • Lai, H. L., & Li, Y. M. (2011). The effect of music on biochemical markers and self-perceived stress among first-line nurses: a randomized controlled crossover trial. Journal of advanced nursing, 67(11), 2414-2424.
  • Huang, C. Y., Chang, E. T., & Lai, H. L. (2016). Comparing the effects of music and exercise with music for older adults with insomnia. Applied Nursing Research, 32, 104-110.
  • Scullin, M. K., Gao, C., & Fillmore, P. (2021). Bedtime music, involuntary musical imagery, and sleep. Psychological Science, 32(6), 985-997.
  • The explanation of heat loss induced by vasodilation is scientifically sound but lacks specific references. Including studies on how pre-sleep baths or showers influence sleep onset would be beneficial.
  • The benefits of warm feet during sleep are mentioned, but the text should cite studies that have found these specific results.

R: We appreciate the reviewer's comments and have addressed them by incorporating additional scientific evidence. We have revised the text to include more robust scientific support for our claims regarding the effects of warm baths and foot temperature on sleep:

  • Haghayegh, S., Khoshnevis, S., Smolensky, M. H., Diller, K. R., & Castriotta, R. J. (2019). Before-bedtime passive body heating by warm shower or bath to improve sleep: A systematic review and meta-analysis. Sleep medicine reviews, 46, 124-135.
  • “Ko and Lee (2018) found that wearing bed socks significantly enhanced sleep quality and reduced sleep onset time in adults sleeping in cool conditions [Y]. A meta-analysis by Jiang et al. (2024) further corroborated that foot thermal therapy can improve various aspects of sleep quality, particularly in older adults [Z]."
  • Jiang CS, Chen KM, Belcastro F. Effects of Temperature, Duration, and Heating Height of Foot Thermal Therapy on Sleep Quality of Older Adults: A Systematic Review and Meta-Analysis. J Integr Complement Med. 2024
  • While the general information about melatonin is accurate, the discussion on its effectiveness, particularly for shift workers, should be backed by studies that have investigated these claims.
  • The text mentions the use of herbs like Valerian and Chamomile but lacks references to scientific studies. Including research on the efficacy of these herbs in improving sleep would be beneficial.
  • The role of tryptophan in serotonin and melatonin production is correctly noted, but the text should reference studies that have examined the impact of tryptophan-rich diets on sleep quality.

R: We appreciate the reviewer's feedback and agree that additional scientific support is necessary. We have revised the text to address these concerns:

  • Liira, J., et al. (2014). Pharmacological interventions for sleepiness and sleep disturbances caused by shift work. Cochrane Database of Systematic Reviews, (8).
  • Shinjyo, N., et al. (2020). Valerian Root in Treating Sleep Problems and Associated Disorders—A Systematic Review and Meta-Analysis. Journal of Evidence-Based Integrative Medicine, 25.
  • Hieu, T. H., et al. (2019). Therapeutic efficacy and safety of chamomile for state anxiety, generalized anxiety disorder, insomnia, and sleep quality: A systematic review and meta-analysis of randomized trials and quasi-randomized trials. Phytotherapy Research, 33(6), 1604-1615.
  • Sutanto, C. N., et al. (2022). The impact of tryptophan supplementation on sleep quality: a systematic review, meta-analysis, and meta-regression. Nutrition Reviews, 80(2), 306-316.